# Database of global glendonite and ikaite records throughout the Phanerozoic

Mikhail Rogov[1], Victoria Ershova[1,2], Oleg Vereshchagin[2], Kseniia Vasileva[2], Kseniia Mikhailova[2], Aleksei Krylov[2,3]

[1] Geological Institute of RAS, Moscow 119017, Russia

[2] Institute of Earth Sciences, St. Petersburg State University, 199034 St. Petersburg, Russia

[3] VNIIOkeangeologia, 190121, St. Petersburg, Russia

*Correspondence to*: Mikhail Rogov (russianjurassic@gmail.com)

**Abstract.** This database of Phanerozoic occurrences and isotopic characteristics of metastable cold-water calcium carbonate hexahydrate (ikaite; $CaCO_3 \cdot 6H_2O$) and their associated carbonate pseudomorphs (glendonites) has been compiled from academic publications, explanatory notes and reports. Our database including more than 700 occurrences reveals that glendonites characterize cold-water environments, although their distribution is highly irregular in space and time. A significant body of evidence suggests that glendonite occurrences are restricted mainly to cold-water settings, however they do not occur during every glaciation or cooling event of the Phanerozoic. While Quaternary glendonites and ikaites have been described from all major ocean basins, older occurrences have a patchy distribution, which may suggest poor preservation potential of both carbonate concretions and older sediments.

The data file described in this paper is available on Zenodo at https://doi.org/10.5281/zenodo.4386335 (Rogov et al., 2020).

## 1 Introduction

Metastable cold-water calcium carbonate hexahydrate (ikaite; $CaCO_3 \cdot 6H_2O$) and their associated carbonate pseudomorphs (glendonites) have attracted considerable attention during the past few decades, mainly due to their possible utility for palaeoenvironmental (especially palaeoclimatic) reconstructions (Kemper and Schmitz, 1975, 1981; Kaplan, 1978, 1979, 1980; Suess et al., 1982, among others). Carbonate pseudomorphs of metastable ikaite have acquired a number of different names (anthraconite, glendonite, thinolite, jarrowite, pseudogaylussite, gennoishi, hokou-seki, White Sea hornlets, polar euhedrons, barleycorn, Gerstenkörner, hedgehogs, see Supplementary), and despite nearly 200 years of study (since the earliest papers by Sokolov, 1825; Freisleben, 1827; Pander, 1830), their distribution and formation mechanisms are still poorly understood. We use the most common name for these pseudomorphs (glendonite), which is derived from their famous locality, Glendon (NSW, Australia, see Dana, 1849; David et al., 1905). Modern ikaite occurs in a wide range of cold-water

environments. Ikaite microcrystals have been recorded in sea ice and cold caves, ikaite tufa has been reported from shallow marine and lacustrine settings, and macrocrystals and their aggregates have been described from hypersaline springs, lacustrine and marine sediments, ranging from littoral environments to abyssal depths (up to ~6,950 m) (Pauly, 1963; Stein, Smith, 1985; Ito, 1996; Dieckmann et al., 2008; Last, 2012; Geptner et al., 2014; Fink et al., 2014; Oehlerich et al., 2015). Occurrences of ikaite pseudomorphs (glendonites) in the geological record are characterized by a slightly less diverse range

of palaeoenvironments. Ancient glendonites have been mainly described from marine clastic rocks, although they have been discovered in caves and lacustrine deposits of Oligocene age (Larsen, 1994).

The first attempts at a comprehensive review of glendonite occurrences through geological time were undertaken by M.E. Kaplan (1978, 1980), who summarized the available information about glendonite findings throughout the world and publicized their importance for palaeoclimatic studies. During the last 40 years, numerous papers on glendonite and ikaite

occurrences have been published, along with new geochemical data. These geochemical data mainly include stable carbon and oxygen isotope values from ikaite and glendonite, along with a limited number of clumped isotope and Sr isotope data (Nenning, 2017; Rogov et al., 2018; Vickers et al., 2020). Information on glendonite occurrences is scattered throughout hundreds of papers on regional geology, explanatory notes to geological maps, and other reports, but until now, no single reference database of global glendonite distribution throughout the Phanerozoic has been compiled.

This paper presents a new single reference database of ikaite and glendonite records from the Early Cambrian until the present. It is based on the analysis of numerous papers (including unpublished reports) and museum collections, as well as on data collected during the last 15 years by the authors. All records were averaged to substages (if possible) without further subdivision, and nearby occurrences were considered as a single locality if the distance between separate data points was less than 1−2 km. As glendonite occurrences were frequently mentioned but not imaged or fully described in many

publications, consideration of such records as true glendonites was mainly based on additional lines of evidence supporting glendonite occurrence within the same region or stratigraphic interval.

## 2 General information about ikaite and glendonite composition

### 2.1 Mineralogy and Petrology

Even though glendonies are known for more than 200 years (e.g. Sokolov, 1825), the precursor mineral (ikaite) was

not discovered until 1963, when Pauly described it from from Ikka (Ika) Fjord, Greenland. Single crystal X-ray structure determinations (e.g., Hesse and Kuppers, 1983; Swainson and Hammond, 2001; Lennie et al., 2004) have shown that ikaite crystallizes in the spacegroup $C2/c$, and its structure consists of $CaCO_3 \cdot 6H_2O$ units with each Ca bound to six water molecules and a bidentate carbonate group to yield 8-fold coordination. Hydrogen bonding links $CaCO_3 \cdot 6H_2O$ moieties to form the crystal structure (Lennie et al., 2004).

Ikaite has been found in natural modern-day environments at temperatures ranging from −2°C to +7°C (Dieckmann et al., 2008; Huggett et al., 2005; Suess et al., 1982). The relative abundance of ikaite in nature is attributed to its

metastability (e.g., Marland 1975). Marland (1975) and Shahar et al. (2005) suggest that low temperatures and high pressures are the main factors controlling ikaite stabilization, and argues that metastable ikaite should not naturally occur on Earth's surface and other carbonates (e.g., calcite, vaterite, aragonite) should crystallize instead. The main factors proposed for ikaite

stabilization at Earth's surface are elevated contents of sulphate/phosphate/magnesium ion in the crystallization medium, or high pH.

Unfortunately, very few studies have investigated the chemical composition of ikaite (e.g., Suess et al., 1982; Pauly, 1963; Schubert et al., 1997). These few studies did not identify sulphate or phosphate ions as constituents of ikaite, whereas magnesium content ranged up to ~0.5 wt.% MgO (Pauly, 1963; Schubert et al., 1997). Schubert et al. (1997) reported minor

amounts of Mg, Fe and Al ($540\pm27\mu g\ g^{-1}$, $43.2\pm1.2\mu g\ g^{-1}$, and $19.4\pm0.2\mu g\ g^{-1}$, respectively) in two dried ikaite subsamples, along with a total organic carbon content of 0.15%. It's important to note that ikaite unit cell parameters provided in published data differ significantly (e.g., Rysgaard et al., 2013), which could indicate chemical composition variations of studied ikaites.

Outside of the aqueous environment, ikaite rapidly disintegrates into a mush of water and anhydrous calcium

carbonate (e.g., Pauly, 1963; Bischoff et al., 1993), which can be in the form of amorphous calcium carbonate (e.g., Zou et al., 2018), vaterite (e.g., Shaikh, 1990; Ito, 1996; Ito et al., 1999; Tang et al., 2009), aragonite (e.g., Stein and Smith, 1986; Council and Bennett, 1993), or calcite (e.g., Pauly, 1963; Bischoff et al., 1993). Ito (1998) suggested that the rate of ikaite transformation is dependent on the availability of water (which increases the rate of transformation) and the presence of magnesium ion (which inhibits the transformation to both calcite and vaterite). Tang et al. (2009) argued that transformation

of ikaite controlled structurally and by kinetic control as the rate of decomposition to vaterite increases with temperature. Purgstaller et al. (2017) showed that the formation of anhydrous calcium carbonates is controlled mainly by the prevailing physicochemical conditions, such as the Mg/Ca ratio of the aqueous medium and water availability. Stockmann et al. (2018) confirmed that and showed that the formation of ikaite is unrelated to the aqueous phosphate concentration. On the other hand, several laboratory experiments have shown that phosphate in solution could have an effect on calcium carbonate

crystallization pathway and consequently on ikaite precipitation (Brooks et al., 1950; Bischoff et al., 1993a; Hu et al., 2014, 2015). Moreover, an increased content of phosphate-ions is often found in pore waters from ikaite-bearing sediment layers (e.g., Kodina et al., 2003).

Ikaite-glendonite transformation begins with ikaite destabilization, resulting in a cloudy (Suess et al, 1982) or whitish (Kodina, 2003) interior of the ikaite crystals. The replacement of ikaite by anhydrous calcium carbonate

($CaCO_3 \cdot 6H_2O \rightarrow CaCO_3 + 6H_2O$) leads to a primary volume loss of 70−80% (Suess et al, 1982; Fairchild et al., 2016) and disintegration, as the density of ikaite is much less than of anhydrous calcium carbonates. However, the primary morphology of ikaite can be preserved due to fast replacement by calcite and cementation (Huggett et al., 2005; Selleck et al., 2007). Early diagenesis of ikaite typically takes place in the sulphate-reduction zone and glendonite can be subsequently altered during burial diagenesis, where it can be replaced by younger carbonate generations or non-carbonate minerals such as silica

(Wang et al., 2017), dolomite (Loog, 1980), pyrite (Rogala et al., 2007) or gypsum (Mikhailova et al., 2019).

Several mineralogical studies have been performed on glendonites (David et al., 1905; Boggs, 1972; Kaplan, 1979; Larsen, 1994; McLachlan et al., 2001; Greinert and Derkachev, 2004; Huggett et al., 2005; Selleck et al., 2007; Vickers et al., 2020). These studies showed that glendonites are composed of 2−4 consecutive carbonates phases, which differ in morphology and chemical composition (Fig. 2). Calcite is the only anhydrous calcium carbonate, which was found as replacive phase after ikaite in natural pseudomorphs (glendonites).

The first carbonate generation comprises blocky calcite crystals, which typically show euhedral, triangular, or bipyramidal shapes, or irregular habits (e.g., Greinert and Derkachev, 2004; Huggett et al., 2005; Qu et al., 2017). Crystals are clear to opaque, with concentric zonation formed by an admixture of clay or organic matter (Huggett et al., 2005) and show no luminescence (e.g., Larsen, 1994; Morales et al., 2017). These crystals are always Mg, Fe and P- depleted (Vickers et al., 2020) or almost pure $CaCO_3$ (Mikhailova et al., 2019). This first generation is interpreted as ikaite-derived calcite (Huggett et al., 2005).

Ikaite-derived calcite does not form the frame of the glendonite and is always supported by cements of different morphology. The first ikaite-derived calcite generation can be overgrown by needle-like or spherulitic calcite cement (e.g., Boggs, 1972; Kaplan, 1979; Huggett et al., 2005; Vickers et al., 2018). Crystals are a yellowish or amber colour under the optical microscope with a bright cathodoluminescence (Frank et al., 2008; Teichert, Luppold, 2013, among others). This carbonates are typically Mg or/and Fe-rich. These carbonates can contain inclusions of pyrite (either as idiomorphic crystals or with framboidal habits; Greinert and Derkachev, 2004). Blocky, sparry or radiaxial fibrous calcite can occupy the residual pore space (Teichert, Luppold, 2013) or replace carbonate rims and ikaite-derived calcite (Vasileva et al., 2019), typically displaying an orange to dark red cathodoluminescence. The last calcite generation is typically Mg / Fe / Sr / P-richest (e.g., Vickers et al., 2020)

Besides multiple carbonate generations, some detrital material is typically also found in glendonites. Inclusions of quartz, feldspar (plagioclase/K-feldspar), volcanic glass, olivine, pyroxene, amphibole, magnetite, hematite, and mica can also occur (e.g., Astakhova and Sorochinskaya, 2000). Glendonites are also tend to have a high organic matter content, which is enough to be measured for stable isotopes by dissolving the carbonate (e.g., Vickers et al., 2020).

## 2.2 Morphology

Unlike calcite and aragonite, ikaite is typically characterized by pyramidal, spear-like (e.g., Dieckmann et al., 2008; Tang et al., 2009; Rysgaard et al., 2014) or stellate crystals (e.g., Selleck et al., 2007) with square-prismatic cross-sections (e.g., Swainson and Hammond, 2001) (Fig. 1). In some cases, the shape of the original ikaite crystal is preserved as a pseudomorph (e.g., Kaplan, 1978; Shearman and Smith, 1985; Kemper, 1987). The size of natural modern ikaite clusters and solitary crystals varies from ~ 5 μm (Dieckmann et al., 2008) to ~12 cm (Kodina et al., 2003). Glendonites display a number of internal structures, including (1) visible core and rim with no apparent zonation; (2) core and alternating mm-scale rims, forming distinct zonation and (3) homogeneous bodies from the centre to the edges. Glendonites are characterized by a size

range which mainly lies between 0.5 and 15−20 cm, with some of the largest specimens ranging up to 1 m in diameter or in length. Microglendonites are rarely reported; however, their size is similar to that of ikaite microcrystals, ranging from 5 to 10 µm (Oehlerich et al., 2013). Glendonite colour varies from light yellow and whitish if weathered to dark brown.

Due to water removal during ikaite to anhydrous calcium carbonate transformation, glendonites are frequently porous and contain small pieces of adjacent sediment (such as sand grains, microfossils, clay particles etc) and several successive carbonate generations.

## 2.3 Isotopic composition of ikaite and glendonite

Stable carbon and oxygen isotopic values of glendonites and ikaites can encompass a broad range of values (fig. 3). $\delta^{13}C$ values of ikaite range from −42.7 to +8.3 ‰PDB, while $\delta^{18}O$ values range from −17 to +3.60 ‰PDB (Dahl, Burchardt, 2006; Chaikovsky, Kadebskaya, 2014; Kodina et al., 2003; Krylov et al., 2015; Last et al., 2013; Lu et al., 2012, Stein, Smidt, 1985; Zabel, Schultz, 2001). The stable isotopic compositions of bulk glendonites and individual carbonate generations differ significantly. $\delta^{13}C$ values of bulk glendonite samples range from −52.4 to +0.6 ‰PDB, while $\delta^{18}O$ values range from −16.6 to +4.8 ‰PDB (Derkachev et al., 2007; Geptner et al., 1994, 2014; Qu et al., 2017; Willscroft, 2013; Vickers et al., 2018; Morales et al., 2017; Teichert, Luppold, 2013, among others). $\delta^{13}C$ values of ikaite-derived calcite range from −27.7 to −14.3 ‰PDB, while $\delta^{18}O$ values range from −5.1 to +1.3 ‰PDB (Frank et al., 2008). Lastly, $\delta^{13}C$ values of secondary carbonate cements range from −20.2 to −3.9 ‰PDB, while $\delta^{18}O$ values range from −23.1 to −6.2 ‰PDB (Frank et al., 2008; Vasileva et al., 2019; see fig. 3). Based on the range of $\delta^{13}C$ values, the source of carbon during ikaite crystallization and ikaite-glendonite transformation was derived from dissolved inorganic carbon (DIC), decaying organic matter or methane seeping through underlying strata.

Although Mesozoic and Cenozoic glendonites are enriched in Sr, they do not record primary $^{87}Sr/^{86}Sr$ ratios of seawater and show enrichment in $^{86}Sr$ compared to coeval marine carbonates (Rogov et al., 2018; author's data).

## 3 General information about ikaite and glendonite database

### 3.1 Data mining

The authors have collected ikaite and glendonite specimens belonging to different stratigraphic intervals from numerous localities over the last 15 years. In addition to our specimens, we also studied glendonites from museum collections and analyzed information about glendonite/ikaite occurrences from numerous publications, using advanced online searches (including all of the different glendonite names mentioned above) along with library searches (see supplementary information for details). The locations of all ikaite/glendonite-bearing sites are shown on Fig. 4. The total amount of records included in the database is 753, based on 376 references, our investigations and communications by our colleagues.

## 3.2 Database contents

The database is provided as an Excel 2003 file (.xls). The main sheet of the database includes the following information: locality, age (substage/formation/Ma), modern coordinates, palaeolatitudes (calculated using paleolatitude.org; see van Hinsbergen et al., 2015), references, depositional settings, and host rock. Some datapoints lack information about palaeolatitude (as they were derived from terrains with an uncertain geographic position) or host rock (in the case of some museum specimens). We aim to provide locality and coordinate information as precisely as possible. However, in some cases the available locality information was rather imprecise (for example, 'basin of river 'X'', or 'mountain ridge 'Y''), although the effects of such imprecise location details on our analysis of (palaeo)latitude distribution are considered insignificant. Palaeolatitudes of Paleogene and Neogene glendonites from the northern margin of the Pacific cannot be determined using paleolatitude.org. We chose a similar palaeolatitude to the present-day for the Paleogene and Neogene based on Bazhenov et al. (1992), Harbert et al. (2000) and Kovalenko and Chernov (2003). Palaeomagnetic data from Sakhalin (Weaver et al., 2003; Zharov, 2005) are more complicated and suggest a northward drift during the Cenozoic. The complex tectonic structure of the island precludes an accurate calculation of palaeolatitude for each time slice and locality from Sakhalin. Database includes also stable carbon and oxygen isotope data and references.

## 4 Data overview

### 4.1 Glendonite distribution in space and time

A significant irregularity in Phanerozoic glendonite distribution is apparent in Fig. 4, which illustrates glendonite locations plotted on present-day geography. However, when the palaeolatitudes of glendonite samples are taken into account, it becomes clear that they mainly formed in high latitudes throughout the Phanerozoic (Fig. 5). Interestingly, most glendonite occurrences have been reported from the northern Hemisphere, which is challenging to explain. Only Quaternary records of ikaite/glendonite occurrences are fairly evenly distributed across the cold-water environments of both the northern and southern Hemispheres and within all ocean basins, including deep-water sites at low and near-equatorial latitudes. Glendonite occurrences within the Arctic Ocean (Kara Sea, Laptev Sea, Chukchi Sea) and the Sea of Okhotsk are more abundant compared to other regions, which can be partially explained by sampling bias, as these regions have been particularly actively studied during the past few decades. In contrast to other geological periods, Neogene glendonites are restricted to the northern margin of the Pacific, except for a single reported occurrence in Arctic Canada. Glendonite occurrences are particularly common at numerous localities along the western coast of the Pacific Ocean, including from Japan, Sakhalin and Kamchatka, where their presence is used as a diagnostic feature for numerous formations and members. Furthermore, one of these formations (Gennoishi Formation of Sakhalin Island) is named after the Japanese name for glendonites (gennoishi, 玄能石). Coeval glendonite records also occur along the eastern Pacific coast, where they were first recognized by Dana (1849). Paleogene glendonites typically occur in the same region as Neogene occurrences but are also abundant in the North Atlantic area. North Atlantic occurrences include giant glendonites from the Mors and Fur islands

(Denmark), as well as abundant Spitsbergen glendonites. It should be noted that the giant glendonites from Denmark are mainly embedded in post-PETM rocks, but clumped isotope data from the glendonites are indicative of near-freezing temperatures (Nenning, 2017; Vickers et al., 2020).

The Late Cretaceous was characterized by a global greenhouse climate and lacks any glendonite occurrences. Lower Cretaceous glendonites are known from all stages and occurred in high latitudes of both the northern and southern
Hemispheres. However, no occurrences are known from the most prominent warming event (early Aptian) and only a single occurrence has been reported from the Barremian. The paucity of Barremian glendonites may be related to the HALIP-induced regression in the Arctic and scarcity of marine deposits of this age in the high northern latitudes. Glendonites are particularly abundant within the Valanginian, Aptian and Albian sediments of Svalbard (Kemper, Schmitz, 1981; Vickers et al., 2018) and Arctic Canada (Herrle et al., 2015), along with the Aptian of Australia (de Lurio, Frakes, 1999). Berriasian
glendonites, which mark the initiation of the Early Cretaceous cold interval, are rare and mainly described from Siberia (Rogov et al., 2017).

Peak glendonite abundance, comparable to that of the Quaternary, occurs during the Middle Jurassic, but nearly all occurrences of this age have been described from northern Eurasia, with no occurrences in the southern Hemisphere. The numerous Middle Jurassic glendonite occurrences coincide with long-term cooling of the Arctic Ocean, primarily instigated
by changes in oceanic circulation following closure of the Viking corridor (Korte et al., 2015). The Late Jurassic is characterized by a significant decrease in glendonite abundance towards the end of this time interval. Lower Jurassic glendonite occurrences are restricted to the upper Pliensbachian and a few in the upper Toarcian.

No glendonites have been confidently identified from the Triassic Period. Only two suspicious reports of pseudomorphs resembling glendonites from low-latitude lagoonal environments were mentioned by Kaplan (1979), based on
reports by van Houten (1965) and Kostecka (1972).

Permian glendonite occurrences display a similar distribution to Lower Cretaceous occurrences. They are also known from both the northern and southern Hemisphere, but are especially abundant in Tasmania and Australia (Selleck et al., 2007), while in the northern Hemisphere they are mainly found in north-east Asia (Biakov et al., 2013), accompanied by rare findings in the rest of Siberia and Novaya Zemlya. Possible glendonites from Turkey and Saudi Arabia are poorly
described.

Carboniferous glendonites are uncommon and mainly known from the Upper Carboniferous of Gondwana. In addition, glendonites were also reported from low-latitude carbonate deposits of Alberta, Canada (Brandley, Krause, 1994).

There is a notable absence of glendonite occurrences for a prolonged period of time below the Carboniferous, spanning the Devonian, Silurian and Middle−Upper Ordovician. Lower Ordovician glendonites have only recently been
discovered and described (Popov et al., 2019; Mikhailova et al., 2019), as until recently, they were known as a kind of 'anthraconite' and were not considered as glendonites. Their findings are restricted to Baltoscandia. Cambrian glendonites are less frequent but are mainly known from the same region and same facies (black shales) as the Lower Ordovician

glendonites. A few Cambrian glendonites were also found in sandstones, and a single occurrence is known from outside the Baltic region in South Korea (Chon, 2018).


## 4.2 Glendonites as palaeoenvironmental indicators

Over the past few decades, glendonites have been used as a proxy for cold-water environments and/or cooling towards glaciation events (Kaplan, 1980; Kemper, Schmitz, 1981; Suess et al., 1982; De Lurio, Frakes, 1999; Price, 1999; Swainson, Hammond, 2001; Selleck et al., 2007; Rogov et al., 2017, among others). This interpretation is grounded on the
discovery of ikaite precipitation during early diagenesis at low temperatures in Holocene−Pleistocene marine sediments, confined to layers with elevated carbonate alkalinity, high pH, and high concentrations of dissolved phosphate (e.g., Kodina et al., 2003; Greinert & Derkachev, 2004; Zhou et al., 2015). The increase in alkalinity, which promotes crystallization of ikaite, is caused by biogeochemical processes in marine sediments, such as organoclastic sulphate reduction and/or anaerobic oxidation of methane (Schubert et al., 1997; Suess et al., 1982; Kodina et al., 2003; Krylov et al., 2015; Lu et al., 2012;
Zabel & Schulz, 2001).

However, experiments results in the last few years suggest at least short-term ikaite stability at much higher temperatures, up to 30-35$^{\circ}$ C  (Clarkson et al., 1992; Rodríguez-Ruiz et al., 2014; Purgstaller et al., 2017; Stockmann et al., 2018; Tollefsen et al., 2020). Ikaite stability at high temperatures had only previously been demonstrated under high pressures (van Valkenberg et al., 1971). It is important to note that high-temperature experiments were only conducted on
ikaite microcrystal precipitates, and there is currently no evidence for stability of 'typical' centimetre to decimetre-sized ikaite crystals and ikaite clusters at similarly high temperatures. Besides that, synthetic ikaites are mainly synthesized in simplified systems, whereas natural ikaites may contain some structural impurities, which could sufficiently influence their field of stability.

Our analysis of glendonite distribution through space and time has revealed a complex relationship between climate
change and glendonite occurrence. On the one hand, nearly all glendonite findings known to date (except for a few doubtful ones) are associated with cold-water environments and are usually found in sediments with high-latitude low diversity faunas and dropstones (e.g., Price, 1999). On the other hand, no glendonites have been discovered from two prominent glacial intervals in Earth's history (the Late Ordovician and Late Devonian), and their pre-Quaternary occurrences are very irregular, changing to the southern Hemisphere in early Paleozoic to mostly northern Hemisphere in Mesozoic and Cenozoic.
However, the roles of additional factors influencing ikaite precipitation remain unclear.

## 5 Data availability

The .xls file containing this database is available on Zenodo  https://doi.org/10.5281/zenodo.4386335  (Rogov et al., 2020). New versions of the database will be also published via Zenodo. The most recent version of will always be accessible

via https://doi.org/10.5281/zenodo.4386335, through direct hyperlink ([http://jurassic.ru/_pdf/glendonites_database.xls](http://jurassic.ru/_pdf/glendonites_database.xls)) or
can be requested directly from the first author of this paper.

## 6 Conclusions

Our new reference glendonite and ikaite database can be used to show that glendonites typically occur in cold-water environments throughout the Phanerozoic, but their distribution is highly irregular in space and time. Although there is no evidence for glendonite formation in warm-water settings, not all of Earth's major glaciations and other cooling events are
associated with glendonite occurrences. Significant irregularity also occurs in spatial glendonite distribution throughout the Phanerozoic. Quaternary glendonites and ikaites have been described from all ocean basins of the world, while pre-Quaternary glendonites have not been found in many regions which would appear to be suitable for ikaite precipitation.

## Author contributions

MR designed the study and collected data on stratigraphic and geographic distribution of glendonite, VE did the
overall editing and supervision, OV collected data on morphology and mineralogical composition, KV and KM were responsible for data on isotopic composition, petrography and cathodoluminescence, AK provided data on ikaite distribution and diagenesis. All authors discussed the results and participated in preparation of the paper.

## Competing interests

The authors declare that they have no conflict of interest.

**Acknowledgements**

The authors thank all colleagues who provided data and samples for this research. We cordially thank Drs. A.R. Sokolov (TsNIGR Museum, Saint-Petersburg, Russia), P.Yu. Plechov (Fersman Mineralogical Museum of RAS, Moscow, Russia) and I.A. Starodubtseva (Vernadsky Geological Museum, Moscow, Russia) for donating specimens of glendonites used in this study. Very important glendonite specimens from the Barents Sea shelf were provided by Dr. M.N. Rudenko
(VNIIOKeangeologiya, Saint-Petersburg, Russia) and A.S. Strezh (VNIGNI, Moscow, Russia). Information about glendonite occurrences (mainly stratigraphic data and coordinates of localities) was received from Dr. A.S. Biakov (North-East Interdisciplinary Scientific Research Institute, FEB RAS, Magadan, Russia), Drs. A.V. Dronov, V.V. Kostyleva, A.B. Kuzmichev, T.N. Palechek, B.G. Pokrovsky, M.I. Tuchkova, V.A. Zakharov (Geological Institute of RAS, Moscow, Russia), Dr. V.M. Gorozhanin (Institute of Geology of USC, Ufa, Russia); Dr. E.A. Gusev (VNIIOkeangeologiya, Saint-
Petersburg, Russia), Dr. A.A. Feodorova (Geologorazvedka, Saint-Petersburg, Russia), Dr. G.S. Iskul (VSEGEI, Saint-

Petersburg, Russia), Dr. M. Jochmann (UNIS, Svalbard, Norway), Dr. V.A. Marinov (Tyumen Oil Scientific Center, Tyumen, Russia), Dr. Morales C. (INGEN, Dijon, France), Drs. A.E. Igolnikov, B.L. Nikitenko, L.G. Vakulenko, P.A. Yan (Institute of Petroleum Geology and Geophysics SB RAS, Novosibirsk, Russia), A.V. Osinzev (Arabica Speleological Club, Irkutsk, Russia), Dr. S. Schneider (CASP, Cambridge, UK), Dr. P.B. Schulz (Museum Salling, Fur Museum, Fur, Denmark),
Drs. R.B. Shakirov and M.G. Valitov (Pacific Oceanological Institute FEB RAS, Vladivostok, Russia), Dr. I.A. Tarasenko (Far East Geological Institute, Vladivostok, Russia), Dr. U. Toom (Tallinn University of Technology, Tallinn, Estonia), and Dr. V.R. Trofimov (Krasnoyarsk, Russia). Glendonite records mentioned in unpublished reports by Rosneft Oil Company (Russia) are used with permission from headquarters of Rosneft-Shelf-Arctica company. Special thanks to James Barnet (Camborne School of Mines) for editing the English. We are very grateful for the constructive reviews of the manuscript by
Dr. Madeleine Vickers and anonymous reviewer.

**Financial support**

VE, OV, KV and KM are grateful for financial support from RFBR, project number 20-35-70012. Investigation of Quaternary ikaites by AK was supported by a grant from the Russian Science Foundation 19-17-00226.

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

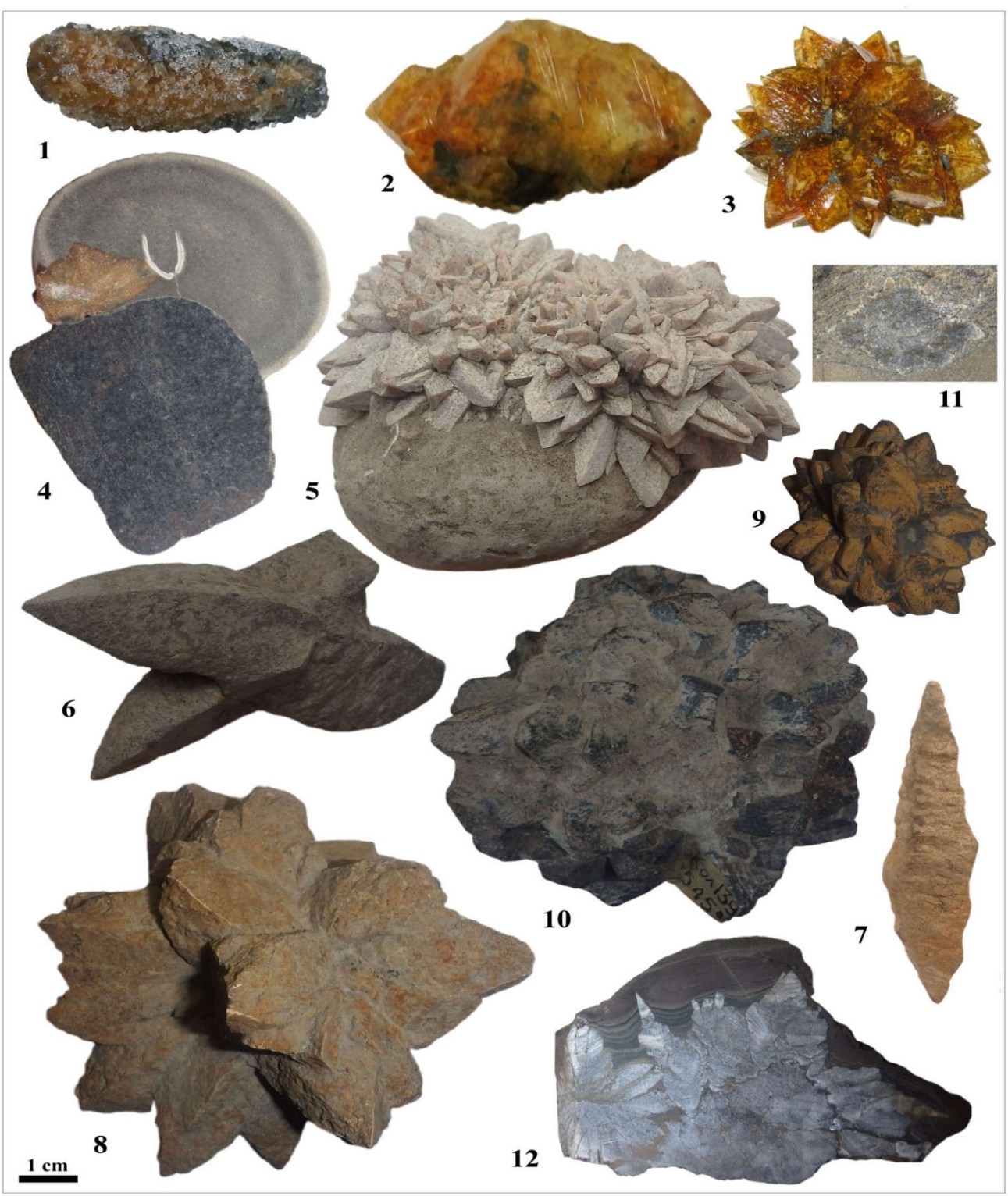

**Figure 1: Ikaite and glendonite morphology (all specimens are figured in natural size). Abbreviation for museum collections: FMM – Fersman Mineralogical Museum, Moscow; GIN – Geological Institute of RAS, Moscow; NHMO – Natural History Museum, Oslo; TsNIGR – Academician F.N. Chernyshev Central Geological Research Museum, Saint-Petersburg; SGM – Vernadsky State Geological Museum, Moscow. 1-3 ikaite; 4-12 glendonite. 1 – Laptev sea, station IP1806, water depth 696 m, level of ikaite sampling 135-137 cm; 2 – Chukchi sea, station 2-2, water depth 50,4 m, level of ikaite sampling 200-230 cm; 3 – Laptev sea, station 12, water depth 31 m, level of ikaite sampling 130-140 cm (Krylov et al., 2015); 4 – specimen embedded in concretion attached to glacial boulder, littoral of the White sea near to Olenitsa river mouth, Holocene (coll. FMM); 5 – glendonite 'twins', Bolshaya Balakhnya river (Taimyr), Quaternary (specimen SGM-276-45/MN-61203, coll. by L.D. Sulerzhitsky, 1981); 6 – south from Pyatibratsky Cape, Western Kamchatka, Eocene (specimen SGM- 445-05/MN-61735, coll. by A.I. Chelebaeva, 1988); 7 – Aralskaya river, Sakhalin, upper Eocene (specimen TsNIGR 4/8561, coll. by Kuzina I.N.); 8 – Tochilinsky section, Western Kamchatka, Eocene-Oligocene transition (specimen GIN, coll. by Palechek T.N.); 9 – Basilika Mt., Spitsbergen, lower Albian (specimen NHMO 924, coll. J.Nagy, 1964); 10 – Olenek river (near Kolumas river mouth), Yakutia, Middle Jurassic, upper Bajocian – lower Bathonian (specimen TsNIGR100/6266, coll. by Gusev A.I., 1939); 11 - Taas-Ary Island, Yakutia, middle Permian, Roadian (field photo by V.B. Ershova); 12 – borehole 60, depth 97,6 m, Saint-Petersburg region, lower Ordovician, Tremadocian (specimen TsNIGR 9/13261, coll. by G.S. Iskul, 2012).**

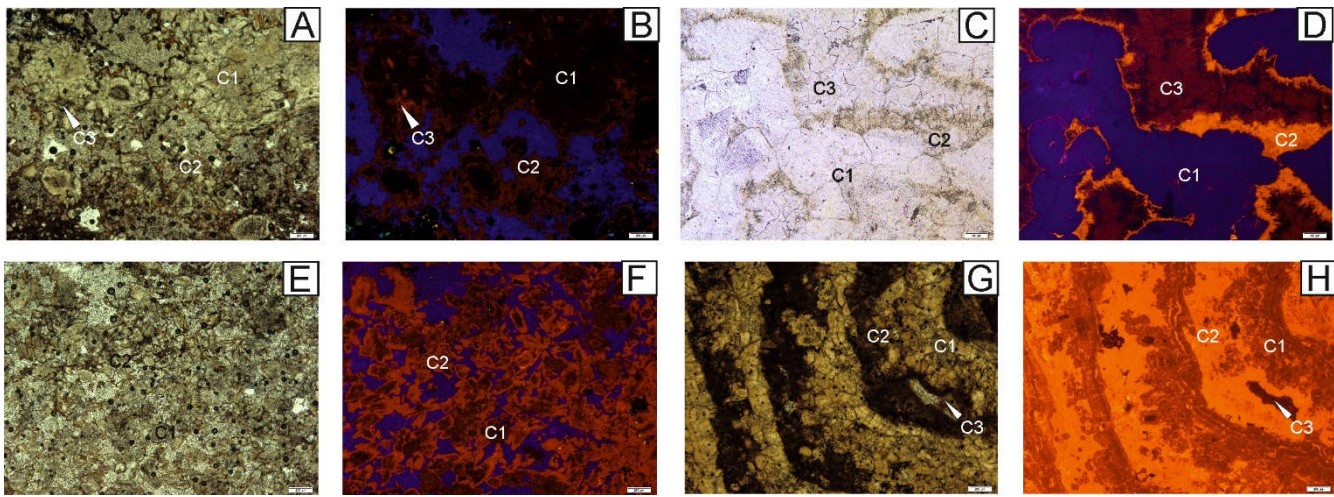

**Figure 2: The internal structure of glendonite that are mainly composed of several calcite phases (C1; C2; C3) A – B. Sample L-1-3, Middle Jurassic (Bajocian-Bathonian), Ludlovskaya-1 well, Barents Sea, Russia, depth 1592+6 meters (authors` data); A - under plane-polarized light; the first calcite phase represents rosette-like crystals showing distinct core; B – the same thin-section under CL. C – D. Sample RM2019-27, Lower Neogene (Lower Miocene), Eastern coast of Sakhalin Island, Russia (authors` data). C - under plane-polarized light; D - the same thin-section under CL. E – F. Sample Led-1-3, Middle Jurassic (Bajocian-Bathonian), Ledovaya-1 well, Barents Sea, Russia,**

**depth 1824+6,6 meters (authors` data); E - elongated calcite crystals (C1), surrounded by the second calcite phase (C2), sample Led-1-3 under plane-polarized light; F - the same thin-section under CL. G - H – Sample F3-1-1, Lower Cretaceous (Upper Hauterivian), Festningen, Svalbard (authors` data); G - microzoning glendonite under plane-polarized light; the first calcite phase is distinguished zoning as well; H - the same thin-section under CL**

560

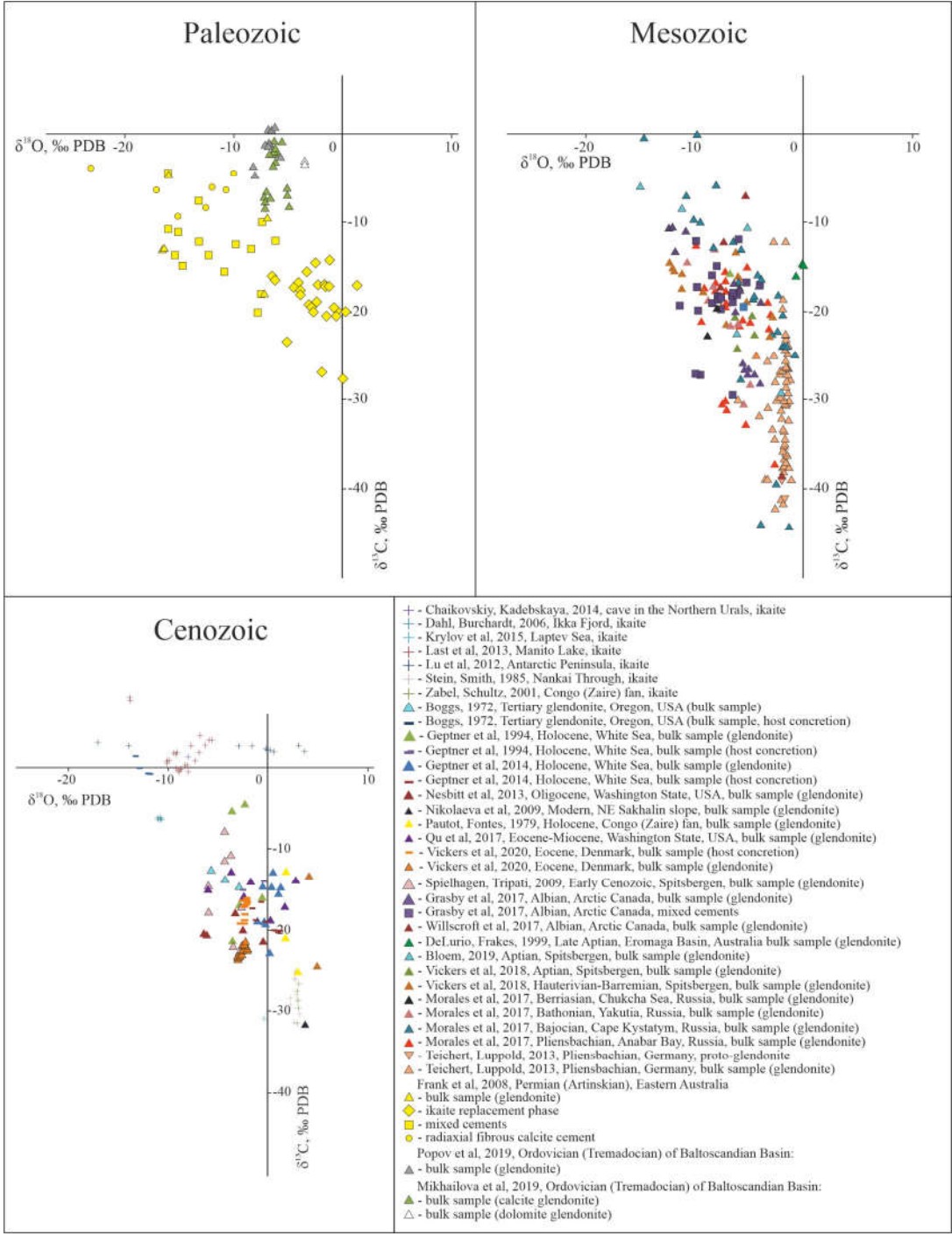

**Figure 3: Isotopic composition of glendonites (including different generations), host concretions and ikaites for Paleozoic, Mesozoic and Cenozoic samples**

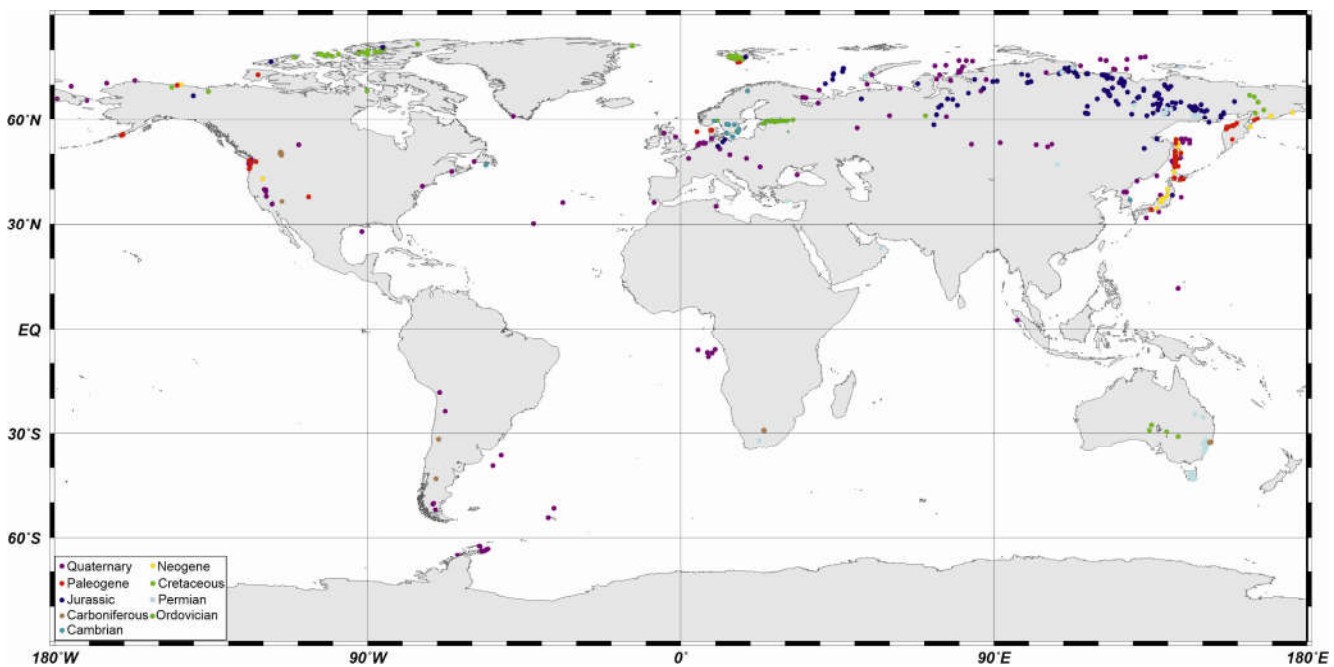

565 **Fig. 4. Map showing location of all ikaite and glendonite occurrences included to the database (prepared through Ocean Data View, Schlitzer, 2020).**

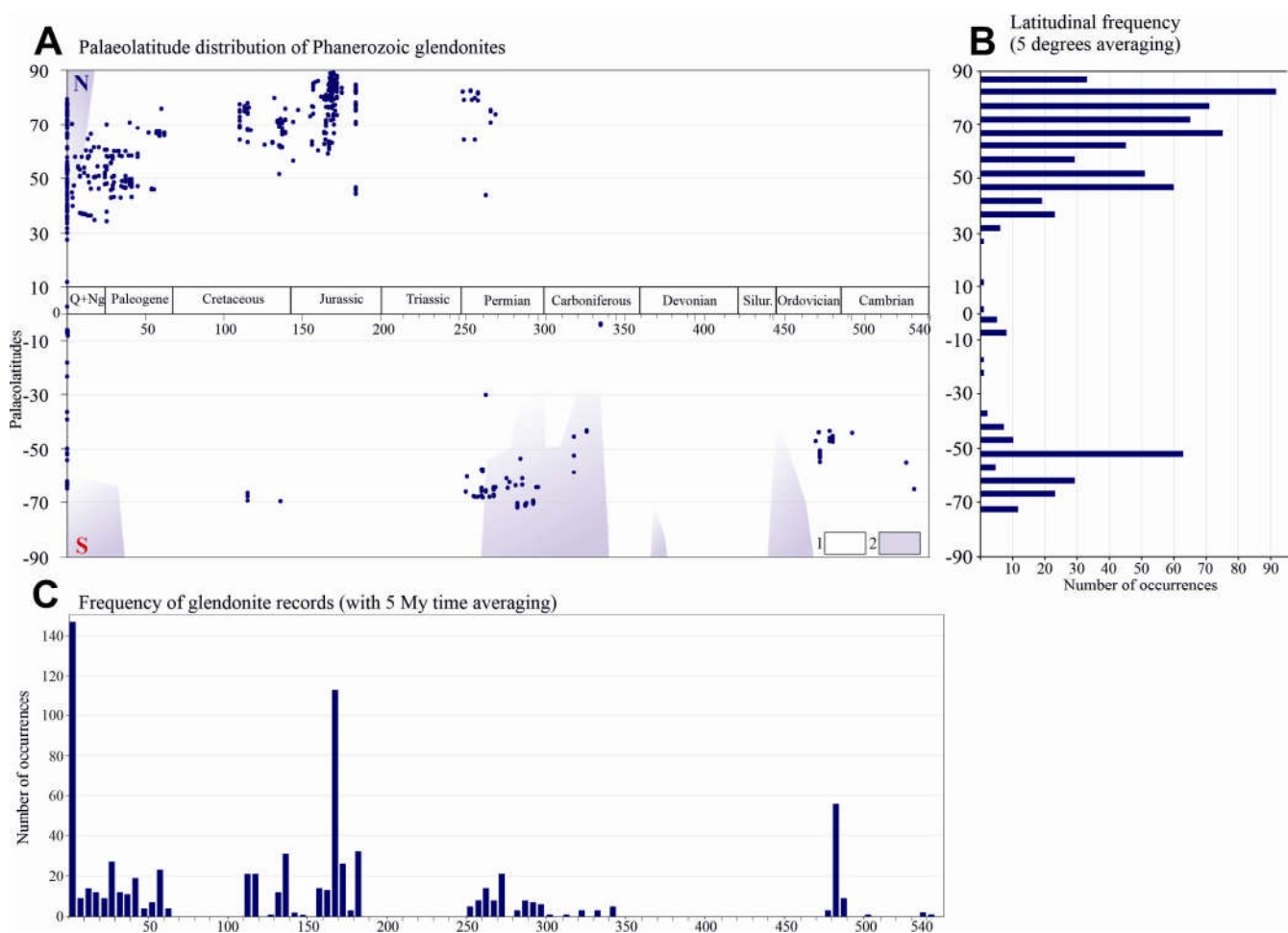

**Fig. 5. Palaeolatitudinal distribution of glendonite and ikaite records through time. A – palaeolatitudinal occurrence;**

570  **1 – non-glacial intervals, 2 - glaciations (after Frakes et al., 2005); B – latitudinal frequency of glendonite occurrences (with 5 degrees averaging); C – frequency of glendonite occurrence through time (with 5 Ma averaging).**