# Peer review of "Database of global glendonite and ikaite records throughout the Phanerozoic"

_Earth System Science Data, 2020_

## Referee Comment (RC1) · Madeleine Vickers (Referee) · 24 Sep 2020

This is a very nice, complete compilation, and has been much-needed by that part of the palaeoclimate community that work on glendonites. There are only a few minor edits I would suggest:

20: "…mainly due to their (possible) utility for palaeoenvironmental (especially palaeo-climatic) reconstructions…" I think the word "possible" should be added as this is still debated among the palaeoclimate community.

70: discussion on transformation from ikaite to water and vaterite, calcite, aragonite….could cite Tang et al. 2009 (J. Applied Crystallography) who also show ikaite transforming to vaterite before the calcite.

[Figure]

75: "Purgstaller et al. (2017) showed that the formation of anhydrous calcium carbonates is controlled mainly by the prevailing physicochemical conditions, such as the Mg/Ca ratio of the aqueous medium and water availability." - Stockmann et al., 2018 (Applied Geochemistry) also show this, and should probably be cited here.

90 – 95: "The size of natural modern ikaite clusters...." Are all of these from personal observation or are there published studies you used to determine this? (in which case, cite them).

100 – 105: Petrography and Cathodoluminescene: You cite studies that have used these techniques: we have just published CL, SEM element maps, and thin section work on the Danish Fur Formation glendonites (Vickers et al., 2020, Nature Communications) that you might want to include here.

120: "Besides multiple carbonate generations, some detrital material is typically also found in glendonites" - They also tend to have a high OM content - enough that can be measured for stable isotopes by dissolving the carbonate (we did this with subsamples of the Danish Eocene glendonites, see Vickers et al., 2020 for the data).

165: "Interestingly, most glendonite occurrences have been reported from the northern Hemisphere, which is challenging to explain." - Could it be partially due to there being fewer high latitude southern hemisphere outcrop studies? Could it be that since many of the Mesozoic and early Cenozoic S. hemisphere studies are based on cores, they may not have sampled rare glendonites in the successions?

180: "It should be noted that the giant glendonites from Denmark are mainly embedded in post- PETM rocks, but clumped isotope data from the glendonites are indicative of near-freezing temperatures" – we also present new clumped and stable isotope data for the Danish Eocene mega-glendonites in Vickers et al. (2020).

180-185: "only a single occurrence has been reported from the Barremian." – Who reported this?

225: However, experiments carried out during the last few years have revealed at least short-term ikaite stability at much higher temperatures, up to 30-35o C" – also Stockmann et al. (2018) could/should be cited here.

Figures: Overall very nice figures. Figure 4, however, the dots showing glendonite/ikaite locations are very small. I think you should make the dots bigger as it's hard to distinguish some of them at this scale (particularly the purple ones)

---

## Referee Comment (RC2) · Anonymous Referee #2 · 18 Nov 2020

The review paper collects comprehensive data of glendonite/ikaite over the Phanerozoic time from worldwide localities. The collection is worth and appropriate to being released by the Earth System Science Data. Before the formal publication, moderate revision is necessary to be done for the MS. MS structureïijŽthe organization of sections is confusing. I suggest an improved structure like that 1. Introduction, 2. Background of ikaite and gelndonite (2.1 Mineralogy and petrology, 2.2 Morphology, 2.2 Isotopic signatures), 3. Method of data compilation (3.1 Data mining, 3.2 Database content), 4. Results and discussion of data compilation (4.1 Temporal and spatial distributions of glendonite, 4.2 Glendonites as palaeoenvironmental indicators), 5 Conclusions. The content of Data availability is presented as Supplementary material in the end of text part). Note: the cathodoluminescence properties should be included in the section

about mineralogy and petrology. The language of the MS needs to be further sharpened. Line 84: cementation and diagenesis occur in sediments related to porewater not seawater. Line110: Mg and Fe-depleted. Line112: cement in different morphology. Line160: Supplementary material. Line171: in contrast to other geological periods. Line181: The occurrence of glendonite over geological time is patchy possibly due to challenges in preservation. Some periods in addition to Cretaceous lack glendonite record. I think it does not mean anything based on the lack of glendonite. It is OK to describe limited existence of glendonite in greenhouse-prevailed periods like Cretaceous. In section 5.1. Temporal distribution of glendonite seems not simple. I suggest the periods can be reorganized into three categories including frequent occurrence, occasional occurrence, and absence. Line 223: experiments results in the last few years suggest…... Line 225-235: I think the discussion should be extended regarding to the paradox between glendonite occurrence and non-low-T settings from both nature and labs. I suggest the discussion can be basically according to temporal and spatial distributions of glendonite summarized in this study. The importance of different environmental factors in glendonite/ikaite formation can be evaluated related to specific scenario and background of glendonite existence.

---

## Author Comment (AC1) · 29 Nov 2020

We are grateful for the helpful review and suggestions by Dr. Madeleine Vickers.

20: ":mainly due to their (possible) utility for palaeoenvironmental (especially palaeoclimatic) reconstructions:::" I think the word "possible" should be added as this is still debated among the palaeoclimate community.

- done

70: discussion on transformation from ikaite to water and vaterite, calcite, aragonite: ::.could cite Tang et al. 2009 (J. Applied Crystallography) who also show ikaite transforming to vaterite before the calcite.

[Figure]

- we cited this article, as well as additional (previously omitted) paper providing evidence of ikaite transformation to amorphous calcium carbonate

75: "Purgstaller et al. (2017) showed that the formation of anhydrous calcium carbonates is controlled mainly by the prevailing physicochemical conditions, such as the Mg/Ca ratio of the aqueous medium and water availability." - Stockmann et al., 2018 (Applied Geochemistry) also show this, and should probably be cited here.

- done

90 – 95: "The size of natural modern ikaite clusters: : :." Are all of these from personal observation or are there published studies you used to determine this? (in which case, cite them).

- proper references were added

100 – 105: Petrography and Cathodoluminescene: You cite studies that have used these techniques: we have just published CL, SEM element maps, and thin section work on the Danish Fur Formation glendonites (Vickers et al., 2020, Nature Communications) that you might want to include here.

- done

120: "Besides multiple carbonate generations, some detrital material is typically also found in glendonites" - They also tend to have a high OM content - enough that can be measured for stable isotopes by dissolving the carbonate (we did this with subsamples of the Danish Eocene glendonites, see Vickers et al., 2020 for the data).

- we cited Vickers et al., 2020 here

165: "Interestingly, most glendonite occurrences have been reported from the northern Hemisphere, which is challenging to explain." - Could it be partially due to there being fewer high latitude southern hemisphere outcrop studies? Could it be that since many of the Mesozoic and early Cenozoic S. hemisphere studies are based on cores, they

may not have sampled rare glendonites in the successions?

- glendonites are so remarkable and easily distinguished from other types of concretions common in sedimentary rocks, that as follow from the both authors' experience and analysis of publications, we rather suggesting that rarity of the Southern Hemisphere occurrences are reflect the real situation, which will be analyzed in detail elsewhere. Although the described database could be partially biased towards the datapoints located in Russia due to availability of diverse data sources, including reports and explanatory notes, online search in different languages indicates rarity of glendonites in the Southern Hemishere except the Australian and Tasmanian occurrences.

180: "It should be noted that the giant glendonites from Denmark are mainly embedded in post- PETM rocks, but clumped isotope data from the glendonites are indicative of near-freezing temperatures" – we also present new clumped and stable isotope data for the Danish Eocene mega-glendonites in Vickers et al. (2020).

- added

180-185: "only a single occurrence has been reported from the Barremian." – Who reported this?

- we found this specimen from the Barents sea shelf in core storage, it will be studied soon. Age of this core was determined by means of dinocysts (Shurekova and Gogin, 2018 – reference added to the database)

225: However, experiments carried out during the last few years have revealed at least short-term ikaite stability at much higher temperatures, up to 30-35o C" – also Stockmann et al. (2018) could/should be cited here.

- added

Figures: Overall very nice figures. Figure 4, however, the dots showing glendonite/ikaite locations are very small. I think you should make the dots bigger as it's hard to distinguish some of them at this scale (particularly the purple ones)

- figure was corrected by changing of grey to light-grey background of landmasses; point sizes were enlarged nearly in two times

[Figure]

[Figure]

**Fig. 1.** corrected fig. 4

---

## Author Comment (AC2) · 29 Nov 2020

We thank anonymous reviewer for valuable comments.

MS structure and the organization of sections is confusing. I suggest an improved structure like that 1. Introduction, 2. Background of ikaite and gelndonite (2.1 Mineralogy and petrology, 2.2 Morphology, 2.2 Isotopic signatures), 3. Method of data compilation (3.1 Data mining, 3.2 Database content), 4. Results and discussion of data compilation (4.1 Temporal and spatial distributions of glendonite, 4.2 Glendonites as palaeoenvironmental indicators), 5 Conclusions.

- proposed structure of the paper is very close to its current one, with minor corrections only. We accepted reviewer's suggestions

[Figure]

Note: the cathodoluminescence properties should be included in the section about mineralogy and petrology.

- corrected

The language of the MS needs to be further sharpened. Line 84: cementation and diagenesis occur in sediments related to porewater not seawater.

- corrected

Line110: Mg and Fe-depleted.

- corrected

Line112: cement in different morphology.

- corrected

Line160: Supplementary material.

- corrected

Line171: in contrast to other geological periods.

- we have revised text following reviewer's suggestions

Line181: The occurrence of glendonite over geological time is patchy possibly due to challenges in preservation. Some periods in addition to Cretaceous lack glendonite record. I think it does not mean anything based on the lack of glendonite. It is OK to describe limited existence of glendonite in greenhouse-prevailed periods like Cretaceous.

- taking into account glendonite occurrences from the diverse terrigenous rocks (and also dissolved holes after glendonites) we are rather suggest that lacking of glendonites from some stratigraphic intervals is corresponding to their real absence or rarity, but not to preservation bias

[Figure]

In section 5.1. Temporal distribution of glendonite seems not simple. I suggest the periods can be reorganized into three categories including frequent occurrence, occasional occurrence, and absence.

- we prefer to consider glendonite distribution in stratigraphic order, as in this case abrupt appearance or disappearance of these pseudomorphs in the geological records became clearer

Line 223: experiments results in the last few years suggest:

- corrected

Line 225-235: I think the discussion should be extended regarding to the paradox between glendonite occurrence and non-low-T settings from both nature and labs. I suggest the discussion can be basically according to temporal and spatial distributions of glendonite summarized in this study. The importance of different environmental factors in glendonite/ikaite formation can be evaluated related to specific scenario and background of glendonite existence

- according to journal rules we provided description of database and brief geological summary rather than long paper with discussion of all features of ikaite/glendonite origin and transformations.

---

## Editor Decision (ED1)

Dear Mikhail Rogov and co-authors,

Many thanks for your very good revision of the manuscript that reads very well!

Before finally accepting the paper for publication in ESSD, I have some questions and a suggestion for your data table

- I am very impressed by your extensive data table and its presentation. However, I wanted to ask you for some adjustments in the URL column of the "references APA" spreadsheet: Please change all URLs leading to DOI referenced articles directly via the DOI link (https://doi.org/10.xxxx/YYYY, e.g. change "https://link.springer.com/article/10.1007/s10347-017-0492-1" in the URL column to https://doi.org/10.1007/s10347-017-0492-1. Using DOI links ensures that the paper will always be reached, even if a URL changes (e.g. when a journal changes the publisher).
- After uploading the new data table to Zenodo, please update the DOI and respective reference in the manuscript (i.e. include the new version and new DOI to the reference).
- Line 253: Please change "Updates will be available under the same web address" (this is not correct, because the DOI above references the actual version 2.0 that should change to 3.0 after you have updated the data table as suggested above) to "New versions of the database will be also published via Zenodo. The most recent version of will always be accessible via https://doi.org/10.5281/zenodo.4289834"
- Line 171: here you are referring to "stable carbon and oxygen isotope data" in the supplementary material. Is this still correct? I didn't find any geochemical data in the Supplementary material. Please clarify!

Many thanks and best regards,

Kirsten Elger

---

## Author Response (AR2)

Dear Kiersten,
Many thanks for your corrections and comments. We improved MS following your proposals.

Before finally accepting the paper for publication in ESSD, I have some questions and a suggestion for your data table
• I am very impressed by your extensive data table and its presentation. However, I wanted to ask you for some adjustments in the URL column of the "references APA" spreadsheet:
Please change all URLs leading to DOI referenced articles directly via the DOI link (https://doi.org/10.xxxx/YYYY, e.g. change "https://link.springer.com/article/10.1007/s10347-017-0492-1" in the URL column to https://doi.org/10.1007/s10347-017-0492-1. Using DOI links ensures that the paper will always be reached, even if a URL changes (e.g. when a journal changes the publisher).
- corrected. A few more references and datapoints were also added, and these updates additionally listed in the new spreadsheet (Additions to v.2)

• After uploading the new data table to Zenodo, please update the DOI and respective reference in the manuscript (i.e. include the new version and new DOI to the reference).
- done

• Line 253: Please change "Updates will be available under the same web address" (this is not correct, because the DOI above references the actual version 2.0 that should change to 3.0 after you have updated the data table as suggested above) to "New versions of the database will be also published via Zenodo. The most recent version of will always be accessible via https://doi.org/10.5281/zenodo.4289834"
- done

• Line 171: here you are referring to "stable carbon and oxygen isotope data" in the supplementary material. Is this still correct? I didn't find any geochemical data in the Supplementary material. Please clarify
"Supplementary include" replaced by "Database includes also"
Our database includes the isotopic data , spreadsheets "Cenozoic glendonite and ikaite", "Mesozoic glendonite", "Paleozoic glendonite"

Mikhail Rogov, on behalf by all co-authors

---

## Author Response (AR3)

Dear Kiersten,

We improved MS following your last correction:
"I am very sorry, but there is still a very small correction required. The DOI in line 254 should be https://doi.org/10.5281/zenodo.4289834 (and not https://doi.org/10.5281/zenodo.4386335). The DOI with the end number 4289834 is that of the collection, which always leeds to the latest version. The DOI you have in the manuscript in line 254 is the one of Version 3.0 (which is the correct DOI for the data described in the manuscript as mentioned in line 252)"
**- corrected**

Best wishes
Mikhail Rogov